# Advancements and Applications of EEG in Gustatory Perception

**DOI:** 10.3390/brainsci15121317

**Published:** 2025-12-10

**Authors:** Lingfeng Yang, Chengpeng Zhang, Wei Wu, Jing Xie, Zhaoyang Ding

**Affiliations:** 1College of Food Science and Technology, Shanghai Ocean University, Shanghai 201306, China; m230351137@st.shou.edu.cn (L.Y.); d250400159@st.shou.edu.cn (C.Z.); jxie@shou.edu.cn (J.X.); 2Shanghai Key Laboratory of Emotions and Affective Disorders, Shanghai Jiao Tong University School of Medicine, Shanghai 201600, China; weiwuneuro@sjtu.edu.cn; 3Shanghai Engineering Research Center of Aquatic-Product Processing & Preservation, Shanghai 201306, China; 4Marine Biomedical Science and Technology Innovation Platform of Lin-Gang Special Area, Shanghai 201306, China

**Keywords:** electroencephalography (EEG), gustatory perception, sensory, brain regions, brain network connectivity

## Abstract

Electroencephalography (EEG) is a powerful tool for investigating gustatory perception, offering high temporal resolution and non-invasive brain activity recording. This review highlights the ability of EEG to reveal the complex interactions between sensory input, emotional responses, and cognitive evaluation in the process of taste perception. This review examines the physiological basis of taste, focusing on key brain regions and how environmental and psychological factors influence taste perception. It also discusses the methods and applications of EEG technology, including its principles, signal features, and measurement methods. Notably, EEG markers like event-related potentials (ERPs), frequency band power analysis, and brain network connectivity are essential for understanding the neural dynamics of taste processing. This review concludes with potential future research directions, including the integration of EEG with other neuroimaging techniques, cross-cultural studies on gustatory perception, and the use of EEG biomarkers in early neurological disease diagnosis.

## 1. Introduction

Gustatory perception plays a central role in how we evaluate and interact with food, influencing our preferences, nutritional choices, and overall enjoyment. This process is complex, involving not only the basic recognition of tastes such as sweet, salty, sour, bitter, and umami, but also the integration of sensory inputs like smell and texture [1]. Beyond sensory processing, gustatory perception is deeply connected to emotions, memories, and cognitive factors. These elements shape how we perceive food and contribute to the variability in individual taste preferences, influenced by genetic, environmental, and cultural factors [2,3]. Recent research has explored these complexities, revealing how taste is not just about the physiological experience but also a multi-dimensional process shaped by various internal and external factors. Advances in neuroimaging technologies, particularly in the context of EEG, have provided new insights into how the brain processes taste and integrates it with emotional and cognitive responses. Notably, taste perception integrates sensory, cognitive, and emotional processes through coordinated large-scale brain networks [4]. Disruption or modulation of these networks directly impacts hedonic evaluation and food selection, underscoring the importance of network-level analysis in understanding taste perception [5].

EEG, phase-contrast magnetic resonance imaging, and electrical impedance tomography are commonly employed in studies of brain taste perception [6]. Among these technologies, EEG stands out due to its high temporal resolution and non-invasive nature, making it an excellent tool for studying gustatory perception [7]. The earliest method explored for EEG recognition was EEG using direct cortical electrodes (ECoG). Although ECoG has been phased out due to its invasiveness, it laid the foundation for neuro-sensory correlation research. Non-invasive EEG has become the mainstream approach for sensory recognition due to its high temporal resolution and non-invasive nature. While non-invasive EEG remains dominant, emerging minimally invasive/semi-invasive interfaces enable precise sensory decoding, hold promise for improved recognition performance, and represent a key focus for future research. EEG measures the brain’s electrical activity in real-time by recording brainwave patterns, which are associated with different cognitive and sensory processes. These brainwaves, categorized into frequency bands such as delta, theta, alpha, beta, and gamma, provide insights into the neural processes involved in taste perception [8,9]. Recent studies using EEG have shown that brain regions such as the insular cortex and the orbitofrontal cortex (OFC) are critically involved in the processing of taste and its integration with emotional responses. For example, a study by Kathrin Ohla et al. [10] demonstrated that the insular cortex plays a pivotal role in both the sensory discrimination of tastes and the emotional evaluation of food, with EEG showing distinct brainwave patterns in response to different tastes like sweet or bitter. EEG is particularly useful for analyzing ERPs, which are time-locked brain responses to specific sensory stimuli. Research by Saša Zorjan et al. [11] found that the P300 component, often associated with attention and cognitive evaluation, was prominently elicited when participants tasted foods, they had positive or negative associations with, highlighting the role of attention and evaluation in gustatory processing. Furthermore, the N400 component, typically linked to semantic processing and emotional responses, was shown to reflect the cognitive evaluation of unfamiliar tastes, signaling how the brain links taste perception with prior knowledge or expectations [12]. Additionally, EEG helps track brain oscillations, revealing how different brain regions interact during taste perception. For instance, studies have shown that theta waves are involved in early sensory processing, particularly in distinguishing between tastes, while alpha waves are linked to higher-order cognitive functions such as decision-making related to food preferences. A study by Diana Rico Pereira et al. [9] demonstrated that alpha band oscillations are prominent when participants are asked to make decisions about food, emphasizing the role of cognitive control in flavor perception and choice. These capabilities make EEG indispensable for understanding the dynamic and multi-dimensional nature of gustatory perception, including its sensory, emotional, and cognitive components.

This work explores the advancements in EEG technology and its applications in gustatory perception research (Figure 1). The first section covers the physiological mechanisms of taste processing, explaining how taste signals are detected and transmitted to the brain, offering a foundation for understanding gustatory perception. The second section examines how environmental, psychological, and individual factors shape taste experience, emphasizing the importance of considering these factors when studying gustatory perception. The third section provides an overview of EEG technology, its principles, and the specific metrics used in food sensory research, highlighting EEG’s unique capabilities for studying gustatory perception. The final section explores the various applications of EEG in gustatory research, including event-related potential (ERP) analysis, frequency band analysis, and brain network analysis, showing how these techniques contribute to a deeper understanding of gustatory processes. This review not only summarizes the current achievements in this rapidly evolving field but also outlines potential directions for future research.

### Literature Search Methodology

To ensure the comprehensiveness and rigor of this review, relevant research was retrieved from PubMed, Web of Science, and Scopus databases using the following keywords: “EEG” OR “electroencephalography” combined with “gustation” OR “taste perception” OR “flavor” OR “event-related potential” OR “neural oscillation” OR “brain connectivity”. The search period was restricted to 2000–2024, focusing on peer-reviewed original research and reviews published in English. Studies were screened based on title and abstract for relevance to EEG-based gustatory research, with full-text review conducted for key publications. A total of 139 studies were included in the final synthesis, covering ERP analysis, frequency band research, brain network studies, and methodological advancements.

## 2. Taste Perception

### 2.1. The Physiological Basis of Taste Mechanisms

Taste, a key human chemosensory function, is critical for nutrient intake and food safety recognition. Its physiological mechanism involves three core links: peripheral signal detection, neural conduction, and central integration [13]. An in-depth understanding of the physiological basis of gustation is important for analyzing human dietary behavior and preventing related diseases. Taste perception begins with the taste bud structures on the tongue and soft palate (Figure 1A). Taste perception is activated when taste stimuli come into contact with taste receptors dissolved in saliva and mucus [14]. These specialized receptor cells are capable of recognizing five basic tastes: sweet, salty, sour, bitter, and umami. The perception of sweet, bitter, and umami is largely dependent on a family of G protein-coupled receptors on the cell membrane [15]. When specific molecules bind to these receptors, they activate a series of intracellular signaling cascades. In contrast, the detection of salty and sour tastes is mediated directly through ion channels, e.g., sodium ions enter the cell through specific membrane channels to trigger depolarization, or hydrogen ions act directly on channel proteins to change their conformation [16]. These different types of sensory information are transmitted via cranial nerves, such as the facial, glossopharyngeal, and vagus nerves, and reach the nucleus tractus solitarius of the medulla oblongata for initial integration before being relayed via the thalamus for projection to higher cortical areas.

In the brain, specific areas of the insula are considered to be the primary taste cortex, which is primarily responsible for the initial analysis and encoding of basic taste characteristics. Here, the nervous system is able to distinguish between different taste qualities and assess their intensity. As information is transmitted to more advanced cortical areas, particularly the anterior insula and OFC, gustatory information interacts in complex ways with higher cognitive functions such as emotion and memory [17]. This integration makes the taste experience not only chemosensory, but also closely related to individual experiences and emotional states. Studies have shown that there are differences in the spatial distribution of sweet and bitter taste representations in the cortex, reflecting the different response patterns of the nervous system when confronted with nutritive and potentially toxic substances [18].

### 2.2. The Influence of Environmental and Psychological Factors on Taste Perception

Taste, as a complex perceptual experience, is not only regulated by a physiological basis, but is also strongly influenced by environmental and psychological factors [3]. The experience of taste in daily life is not only a chemical sensation, but also involves sophisticated sensory integration. Olfactory, tactile and other sensory channels work together to build a complete “flavor” perception. For example, the volatile aroma molecules of a food are integrated with taste bud stimuli, and the texture and temperature of a food influence the taste experience via the trigeminal nerve, which is a key component of taste perception [19].

The modulation of taste function is diverse. Endogenous factors include the regulation of taste sensitivity by metabolic hormones such as leptin and the role of dopamine in the reward mechanism, while exogenous factors involve dietary habits and taste adaptation triggered by environmental factors. Sodium chloride is transmitted through two primary pathways: the aminopyridine-sensitive and aminopyridine-insensitive pathways. The aminopyridine-sensitive pathway is mediated by the epithelial sodium channel (ENaC), through which sodium ions enter taste receptor cells (TRCs) during chewing, triggering depolarization (Figure 1B). This depolarization activates voltage-gated neurotransmitter release channels composed of CALHM1 and CALHM3. These channels open during membrane depolarization, leading to ATP release, which then activates taste afferent neurons. The electrical signals from salt taste receptors are transmitted to the central nervous system, allowing for the perception of saltiness [20]. Long-term high-salt diets reduce the sensitivity to salty taste.

Environmental factors are crucial in taste modulation, especially the effect of visual stimuli on taste anticipation. Studies have shown that food color significantly alters the perception of sweetness and acidity, with red enhancing sweetness and green increasing sourness anticipation [19]. Light intensity also affects taste sensitivity, with flavor discrimination being stronger in bright light [21]. Tactile factors are equally important, with the weight, material and color of utensils systematically altering food taste perception. For example, heavy cutlery increases ratings of food richness, and white plates enhance sweetness perceptions of desserts [22]. This cross-sensory integration shows that taste perception is the result of multimodal construction.

Psychological states play a significant role in taste modulation. Acute stress conditions resulted in reduced salivary secretion, elevated taste thresholds, and diminished sweet and fresh flavor perception. Chronic stress remodeled taste preferences and increased craving for high-fat and high-sugar foods by altering hypothalamic-pituitary-adrenal axis function [23]. Depressed mood reduces taste sensitivity and may be associated with dysfunction of the monoamine neurotransmitter system. The decline in taste function in the elderly is partly due to changes in neuroplasticity caused by reduced environmental stimuli [24]. The multidimensional mechanisms of taste modulation, linking physiology, psychology and the environment, provide a unique window into human eating behavior.

### 2.3. Individual Differences in Taste Perception

Individual differences in taste perception originate from the complex interaction of genetic, molecular and environmental factors, and the analysis of its mechanism and industrial application has become a research frontier in food science and sensory biology. At the molecular level, polymorphisms in taste receptor genes form the core basis of differences. For example, the bitter taste receptor encoded by the TAS2R38 gene exhibits a bimodal distribution of sensitivity to phenylthiourea (PROP), and a single nucleotide polymorphism (SNP) resulting in a proline-to-alanine substitution at position 49 significantly alters the conformation of the receptor’s ligand-binding domain, resulting in about 25% of the population being “hypersensitive” [25,26]. A similar mechanism exists in the sweetness receptor TAS1R2/T1R3 heterodimer, where a mutation at rs35874116 reduces the receptor’s affinity for natural sugars but increases its sensitivity to artificial sweeteners, such as aspartame, and the dynamics of this receptor-ligand interaction directly affects an individual’s threshold of preference for sweet substances [27,28]. In addition, advances in salivary proteomics have revealed that gustin protein (CA6 gene product) regulates taste bud cell differentiation through zinc ion transport, and that its genetic polymorphisms lead to up to three-fold differences in salivary zinc concentration, significantly affecting the density of bacillary papillae and taste acuity [29].

The cascading effects of the above molecular networks are manifested at the macrobehavioral level in significant sensory phenotypic differentiation. PROP hypersensitive individuals have a 40–60% lower aversion threshold to glucosinolate metabolites from cruciferous plants, and this bitter taste avoidance behavior is negatively correlated with vegetable intake, which may affect the composition of the gut flora and the risk of metabolic diseases [30]. Carriers of the sweet taste receptor variant had a 15–20% elevated threshold of pleasurable response to sucrose, but showed hypersensitivity to artificial sweeteners, a perceptual difference that led to a preference for sugar-substituted foods, but may trigger an energy compensation effect [31]. Notably, the oral microbiome is involved in taste modulation through metabolic interventions; for example, *Streptococcus* spp. can reduce nitrate to nitrite and alter the activity of ion channels in the tongue epithelium, whereas certain strains of *Clostridium* spp. can catabolize sulphur-containing amino acids to generate volatile hydrogen sulphide, which indirectly enhances the activation efficiency of bitter taste receptors [32].

The industry has developed precision solutions based on this. The food industry uses Genome-wide association studies (GWAS) to screen SNP loci associated with taste and design graded formulations for different haplotype groups, for example, developing glucosinase inhibitors to reduce the bitter taste of vegetables for individuals carrying the TAS2R38 mutation [32]. In the pharmaceutical field, the use of nanoliposome encapsulation to mask the bitter components of drugs such as ibuprofen has been designed based on the simulation of the kinetics of hTAS2R10 receptor activation. The nutrition industry has integrated multi-omics to establish a taste-metabolism association model, designing low-sugar, high-umami dietary plans for individuals at high risk of diabetes. This approach leverages the high sensitivity of the umami receptors T1R1/T1R3 to monosodium glutamate to compensate for the sensory satisfaction of taste (Figure 1C).

Cutting-edge research is currently making breakthroughs from multiple dimensions, challenging existing scientific paradigms. Based on AlphaFold2, the structural prediction of taste receptors combined with molecular dynamics simulations has successfully elucidated the allosteric binding mechanism of sweeteners with the T1R2/T1R3 receptors, guiding the design of innovative sweet molecules that selectively activate specific receptor subtypes [33]. In the field of neuroregulation, studies have found that transcranial direct current stimulation (tDCS) can temporarily enhance the response of the insular cortex to umami signals, increasing umami intensity perception by more than 30%, providing a new strategy for the intervention of anorexia nervosa [34]. In synthetic biology, efforts are underway to engineer oral symbiotic bacteria to express bitter receptor antagonistic peptides or sweet-enhancing proteins, achieving dynamic regulation of taste phenotypes through microbiome-host interactions [35]. These groundbreaking advancements not only deepen the theoretical framework of taste biology but also signal the industrial prospects of personalized sensory experience customization, driving the food, medical, and health industries into a new era of precision intervention.

**Figure 1 brainsci-15-01317-f001:**
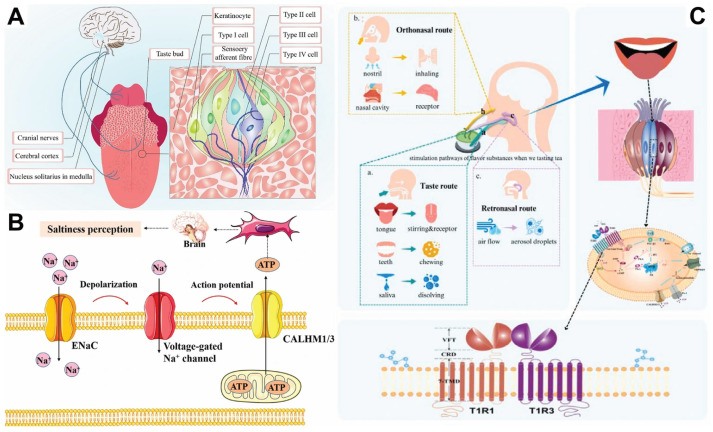
(**A**) Different sensitivity taste maps of the tongue [36]. (**B**) Amiloride-sensitive pathway in taste receptor cells [20]. (**C**) Taste perception triad: anterior nasal cavity, posterior nasal cavity, and taste pathway: a. gustatory pathway, b. orthonasal pathway, c. retronasal pathway [37].

## 3. Technical Characteristics and Sensory Research of EEG

### 3.1. Basic Principles of EEG

Electroencephalography (EEG) is a non-invasive technique for recording the electrical activity of cortical neurons in the brain, with its core principles spanning neurophysiology and biomedical engineering [38]. In 1929, German psychiatrist Hans Berger successfully captured brain electrical signals on the human scalp for the first time—this discovery not only confirmed the observability of brain electrical activity but also marked the beginning of exploring brain functions via electrophysiological methods.

Over nearly a century of development, EEG technology has evolved from initial single-channel recordings to a multimodal neuroimaging tool with millisecond-level temporal resolution, playing an irreplaceable role in clinical neurology, cognitive science, and brain–computer interfaces. Its cortical electrical activity originates from the coordinated work of approximately 14 billion neurons: when neurons transmit information via synapses, the opening of ion channels on the postsynaptic membrane triggers the spatiotemporal integration of local potentials [39]. Among these, pyramidal cells, due to the directional consistency of their dendritic arrangement, are the primary contributors to EEG signals [40]. The postsynaptic potentials generated by the firing of such cell populations are transmitted through the skull-scalp medium by volume conduction, ultimately forming a weak electric field on the scalp surface with an amplitude of 5–100 microvolts. According to the quasi-static electromagnetic field theory, the propagation of these electrical signals follows a simplified form of Maxwell’s equations, with their attenuation closely related to the conductive properties of cerebrospinal fluid, skull, and scalp. The high electrical resistance of the skull (approximately 0.0042 S/m) results in a signal attenuation of more than 90%.

Modern EEG adopts the international 10–20 electrode standard [41], using 19–256 silver chloride electrodes (3 cm^2^ density) to cover the whole brain, based on precise proportional division of cranial anatomical landmarks (Figure 2A). For signal acquisition: an instrumentation amplifier with CMRR > 110 dB eliminates environmental interference; a 0.5–35 Hz bandpass filter removes baseline drift and EMG noise; microvolt-level signals are digitized via a 24-bit analog-to-digital converter.

Raw EEG often has physiological artifacts (e.g., 100–200 μV eye movement signals [42]) and environmental interference (e.g., 50 Hz power noise), which require noise reduction via ICA or adaptive notch filtering [43].

### 3.2. EEG Signal Characteristics and Measurements

EEG signal analysis centers on understanding its physical properties and physiological significance, with frequency features being key to distinguishing neural activity states (Figure 2B). Delta waves (0.5–4 Hz, deep sleep) reflect cortex-thalamus synchronized firing; theta waves (4–8 Hz, light sleep/meditation) link to hippocampal memory [44]. Alpha waves (8–12 Hz) are most prominent during relaxed wakefulness with closed eyes, especially over the occipital lobe, and immediately diminish upon eye opening, a phenomenon referred to as “Alpha block” [45,46]. Beta waves (13–30 Hz), active in the frontal lobe and motor cortex, indicate increased cognitive engagement and are observed during higher-order processing tasks. Gamma waves (30–100 Hz), involving coordinated activity across multiple brain regions, serve as neural markers for perceptual integration [47]. Due to their small amplitude (typically < 10 μV), precise detection of gamma waves requires high-quality electrodes and amplifiers.

In terms of signal amplitude, normal EEG activity typically ranges between 10–100 μV. However, during epileptic seizures, the amplitude can escalate to millivolt levels, reflecting pathological neural activity [48]. Neural firing synchronization directly correlates with EEG amplitude—greater synchronization enhances electrical field summation. Precise EEG measurement requires proper electrode selection: silver/silver chloride (Ag/AgCl) electrodes (stable ~0.222 V potential) are the gold standard, with impedance needing to be <5 kΩ to ensure signal fidelity. For the international 10–20 electrode system, inter-electrode distance is ~10% of head circumference, enabling cross-laboratory data comparability. Additionally, 256-channel high-density EEG improves spatial resolution to ~5 mm, though this remains lower than fMRI’s millimeter-level precision [49].

A complete EEG system employs a three-stage signal processing chain: the front-end amplifier requires a common-mode rejection ratio (CMRR) of over 100 dB to eliminate environmental interference, with a typical input impedance of 1 GΩ; analog-to-digital conversion (ADC) uses 24-bit chips and a sampling rate of 200–1000 Hz for high-resolution signal capture [50,51]. And in subsequent signal processing, a 0.5 Hz high-pass filter is first used to remove baseline drift, followed by a 50/60 Hz notch filter to suppress power-line interference.

In EEG signal analysis, time-domain analysis often utilizes ERPs, with components such as P300 being particularly informative [52]. Changes in the latency of P300 are indicative of cognitive dysfunction. Frequency-domain analysis, on the other hand, uses Fourier transform to calculate the power spectral density of EEG signals, providing insights into brain activity across different frequency bands [53]. In recent years, time-frequency analysis methods, such as wavelet transforms, have gained prominence for capturing non-stationary features in EEG, which is crucial for real-time monitoring and dynamic brain activity analysis.

Despite significant progress in EEG signal analysis, several challenges remain in practical applications. For instance, EMG artifacts in the 20–300 Hz range overlap with the gamma band, requiring surface electromyography (sEMG) reference signals for effective artifact removal [54]. Movement artifacts are particularly pronounced in mobile EEG systems, with the overlap between movement-induced interference and brainwave frequencies posing a significant challenge [55]. Recent solutions have included motion compensation algorithms assisted by inertial measurement units (IMUs) to adjust for movement-related artifacts and enhance signal quality [56].

In clinical diagnostics, automated epileptiform discharge detection has over 90% sensitivity, though false positives remain a key issue. Brain–computer interface (BCI) systems require even higher signal quality—typical SSVEP-based BCI systems achieve an information transmission rate of 60 bits/min [57]. Yet, as application demands grow, further system performance improvements are needed.

Future directions in EEG research include the development of flexible dry electrode technology, wireless data acquisition systems, and integration with other modalities such as functional near-infrared spectroscopy (fNIRS) [58]. These advancements are poised to propel EEG technology from laboratory settings to everyday monitoring applications, opening up new opportunities in remote health monitoring, smart homes, and brain health management.

### 3.3. EEG Indicators Applicable to Food Sensory Research

In sensory food research, EEG provides objective evidence for understanding consumers’ neural cognitive responses to food. It captures the brain’s electrophysiological reactions when participants taste different foods. These responses reflect true preferences, reveal hidden biases and deep sensory mechanisms, and offer a more accurate, objective sensory evaluation tool than subjective self-reports.

When tasting food, the brain produces specific electrophysiological responses in both temporal and frequency domains (Figure 2C). Within 100–300 ms of taste stimulus presentation, early ERP components (N1, P1) emerge first, closely tied to rapid processing in the primary taste cortex [59]. Studies show sweet stimuli often increase P1 amplitude, while bitter stimuli enhance N1; these differences reflect the brain’s ability to distinguish taste qualities and their associated neural responses, providing key neurophysiological data on how taste stimuli quickly trigger brain perception.

As the tasting process continues, the P300 component typically emerges within the 300–500 millisecond time window, which is closely associated with the allocation of cognitive resources. In blind taste tests, foods preferred by participants tend to evoke larger P300 amplitudes, suggesting that the brain allocates more cognitive resources to these stimuli [60]. This finding not only provides evidence of the neural basis for food preferences but also highlights the critical role of attention in the sensory evaluation process.

Furthermore, the late LPP (Late Positive Potential) component typically emerging around 700 ms is closely tied to emotional experience intensity. Pleasant food experiences usually boost LPP amplitude, mirroring emotional responses to food; this component is widely used in emotion research to clarify how foods trigger emotional resonance and shape consumer choices [61].

Meanwhile, when tasting experience contradicts expectations (e.g., a low-sugar drink tasting less sweet than anticipated), a distinct FRN (Feedback Related Negativity) component appears within 200–350 ms. As a negative wave reflecting the brain’s rapid evaluation of prediction errors, FRN not only reveals how the brain processes expectation-actual experience discrepancies but also provides neurophysiological evidence for studying expectation’s role in food selection [62].

Complex taste perception, such as that elicited by multi-component dishes, is not a mere summation of basic taste modalities (sweet, sour, salty, bitter, umami) followed by generalization, but a dynamic neural integration process. EEG studies have revealed that while basic tastes activate relatively distinct cortical regions (e.g., insula, orbitofrontal cortex), complex taste stimuli trigger synergistic activation of these regions plus additional neural networks involved in sensory integration, memory, and context processing. For instance, the interaction between sweet and umami in savory dishes modulates theta and gamma band oscillations in the insula, reflecting non-linear neural encoding beyond simple additive processing. This integration enables the perception of unique “gustatory gestalts” (e.g., the umami-rich complexity of broth or the balanced sweetness-sourness of fruit sauces) that cannot be replicated by individual basic tastes. Such findings highlight the sophistication of complex taste perception and underscore the value of EEG’s high temporal resolution in decoding the dynamic neural mechanisms underlying this process.

In frequency domain analysis, the activity of the Alpha band is significantly associated with emotional states. Research shows that when participants experience pleasant taste stimuli, particularly in the prefrontal cortex, the power of Alpha waves in the left hemisphere significantly decreases [63]. This asymmetrical pattern is considered a neural marker of positive emotional responses. Conversely, unpleasant tastes may lead to an increase in Alpha waves, reflecting the neural mechanisms of negative emotional responses [64]. These findings provide strong evidence for the complex relationship between emotion and taste perception.

Gamma band activity is also associated with higher-order cognitive processing [65]. Stronger gamma oscillations occur when tasting foods with complex or innovative flavors, reflecting the brain’s integration of novel, complex sensory inputs. Such flavor complexity and novelty drive the brain to allocate extra cognitive resources, boosting gamma activity, this offers key insights for exploring flavor complexity’s role in food development and consumer preference research [66].

Beta band activity also matters for sensory food experiences. Studies show tasting foods with rich flavor layers significantly enhances beta waves, signaling the brain’s need to mobilize more cognitive resources to process complex sensory information. This underscores how food complexity affects cognitive resource allocation, providing theoretical support for sensory complexity research in food design [67].

Meanwhile, an increase in Theta waves may reflect cognitive conflict during decision-making processes, particularly in situations involving the dilemma between health and taste [68]. For example, when consumers face the trade-off between healthiness and tastiness, an increase in Theta wave activity may indicate the brain’s processing of the conflict between emotional and rational decisions.

By integrating these temporal and frequency EEG markers, researchers can build more comprehensive neurocognitive models to predict actual consumer choice behavior. The current research trend is to develop more convenient EEG acquisition systems, such as wireless dry electrode devices, which make it possible to conduct research in real-world consumption environments [69]. At the same time, the introduction of machine learning methods has greatly improved the efficiency of EEG signal analysis, allowing for more accurate classification of consumer preferences and behavior patterns.

In the future, the integration of multimodal neuroimaging technologies, including EEG and other brain imaging techniques such as fMRI and near-infrared spectroscopy (NIRS), will further deepen our understanding of food sensory experiences. This multidimensional approach will provide more scientific and systematic evidence for the food industry’s product development, consumer behavior prediction, and marketing strategies, thereby promoting innovation and development in the food sector.

**Figure 2 brainsci-15-01317-f002:**
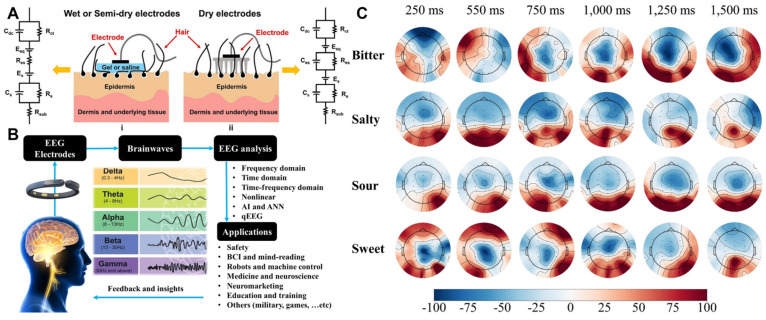
(**A**) Equivalent circuit model of electrode-skin interface for different electrodes [70]. (**B**) Integrative framework of EEG technology from electrode placement to diverse applications [71]. (**C**) Time-domain level information of the brain to the different taste stimuli [72].

## 4. Application of EEG in Taste Research

### 4.1. Analysis of ERPs Induced by Taste Stimuli

ERPs represent finely timed neural responses to sensory, cognitive, or motor events, captured through EEG. These voltage fluctuations provide millisecond-level resolution, making them invaluable for dissecting the temporal dynamics of brain processes. Unlike other neuroimaging techniques, such as fMRI or PET scans, ERPs offer exceptional temporal precision, allowing researchers to track brain activity as it unfolds in real time [73]. This temporal accuracy is crucial for understanding the brain’s processing of complex stimuli, such as taste, which involves rapid, dynamic neural interactions. ERPs are characterized by their latency (the time after stimulus onset), amplitude (signal strength), and polarity (positive or negative deflection). P1, N1, P2, and P300 reflect different stages of cognitive processing and provide key insights into the neural mechanisms underlying perception, attention, and memory (Table 1) [74].

The application of ERPs to gustatory stimuli offers unique insights into taste perception, a process inherently involving multisensory integration. Flavor arises not only from taste but also from the interplay of smell, texture, and even visual cues. This makes the study of gustatory perception more complex than other sensory modalities such as vision or hearing, where sensory processing tends to be more isolated [86]. Gustatory perception involves a wide range of neural and cognitive processes, from basic sensory detection to higher-order evaluations of pleasure or aversion. The ERP method provides an effective means of dissecting these processes in detail, capturing the temporal dynamics of how taste is perceived and evaluated in real time [74].

Gustatory ERP research requires specialized methodologies due to the nature of taste stimuli. Unlike visual or auditory stimuli, which are easily presented through images or sounds, taste stimuli must be delivered using precise delivery systems to ensure consistent timing and volume. Gustatometers are devices designed for this purpose, allowing for the controlled administration of taste stimuli [87]. These devices are capable of delivering a wide range of taste qualities, including sweet, salty, sour, bitter, and umami, while also regulating the concentration of each stimulus [88]. The precision of gustatometers is essential for maintaining the consistency of experimental conditions, which is crucial when studying the neural responses to taste.

One commonly used experimental paradigm in gustatory ERP research is the oddball paradigm, in which a rare taste stimulus (e.g., sweet) is presented among more frequent ones (e.g., salty) [73]. This design allows researchers to examine how the brain processes unexpected or novel stimuli. Oddball paradigms are effective for studying attention and cognitive processing because they elicit a strong response from the brain, particularly in components like the P300, which is associated with attention and memory updating [89,90]. In addition to the oddball paradigm, researchers often incorporate rinsing with water between trials to prevent sensory adaptation. This is important because repeated exposure to the same stimulus can lead to a decrease in the brain’s response to that stimulus, which could confound the results.

Gustatory ERPs unfold in a predictable sequence, with each phase reflecting different neural operations. The early components, occurring within 200 milliseconds of stimulus presentation, are thought to correspond to the initial sensory encoding of the taste. These components, including the P1 (50–100 ms) and N1 (100–150 ms), originate in the primary taste cortex, which includes regions such as the insula and the frontal operculum [74]. The P1 and N1 components reflect the brain’s early detection and categorization of taste stimuli. The N1, in particular, is sensitive to the categorization of basic tastes, such as sweet, salty, or bitter, and is thought to reflect the brain’s initial classification of the stimulus.

As the ERP response progresses, mid-latency components, occurring between 200 and 400 milliseconds, become more prominent. These components, including the P2, are thought to correspond to the hedonic evaluation of the taste. The P2 component is particularly sensitive to the emotional or evaluative aspect of taste, with pleasant tastes (such as sweetness) eliciting larger P2 amplitudes than aversive tastes (such as bitterness) [91]. This suggests that the P2 component may be linked to reward processing in the brain, particularly in areas such as the OFC, which plays a key role in the evaluation of rewards and the hedonic aspects of sensory experiences [92]. Thus, the P2 component provides important information about the subjective experience of taste, as it reflects how the brain assigns value to a particular taste stimulus.

Later components of the ERP, including the P300 (300–600 ms) and LPP, reflect higher-order cognitive processes such as attention, memory updating, and subjective pleasantness judgments. The P300 component, in particular, is associated with attentional processes and cognitive evaluation [93]. It is often used as a marker for assessing the brain’s response to unexpected or novel stimuli, making it particularly useful in oddball paradigms. The LPP, which occurs later in the ERP response, is thought to reflect more sustained processing, such as emotional or evaluative judgments, and may be linked to conscious awareness of taste [94,95].

Several factors modulate gustatory ERP responses, making it essential to consider various aspects of stimulus delivery and individual differences. Stimulus properties, such as concentration and taste quality, have a direct impact on both amplitude and latency of ERP components. For example, bitter stimuli tend to evoke larger N1 and P2 responses due to their innate aversiveness [96,97]. This heightened response is thought to reflect the brain’s increased sensitivity to potentially harmful or unpleasant tastes. On the other hand, higher concentrations of taste stimuli generally amplify all ERP components, as the brain’s response to the stimulus is intensified with increased sensory input.

In addition to stimulus properties, individual differences also play a crucial role in shaping gustatory ERP responses. Genetic variations, such as those seen in supertasters, individuals with heightened taste sensitivity, are associated with enhanced P2 amplitudes [96]. Supertasters have more taste buds and may experience flavors more intensely than non-supertasters. This heightened sensitivity is reflected in the larger P2 responses to pleasant tastes, such as sweetness [98]. Age-related changes in taste processing also influence ERP responses, with older individuals often showing reduced P3 responses to taste stimuli. This may reflect developmental shifts in taste preference, as aging can lead to changes in the neural systems that govern taste perception [99].

Clinically, gustatory ERPs have proven to be a valuable tool for studying neurological and psychiatric conditions. For example, individuals with anorexia or obesity often exhibit altered P3 responses to sweet stimuli, suggesting dysregulated reward processing [94,100]. The P3 component, which is related to the brain’s evaluation of stimuli and reward, can be used to assess how these individuals process hedonic aspects of taste. In Alzheimer’s disease, attenuated P1/N1 components may signal early gustatory dysfunction, as these individuals may have difficulty detecting and categorizing tastes [101]. By examining these ERP components, researchers can gain insight into the neural underpinnings of taste perception in various clinical populations.

Beyond clinical applications, the food industry has also recognized the potential of gustatory ERPs for predicting consumer preferences. Changes in ERP amplitude, particularly in components like the P2 and P300, have been shown to correlate with hedonic evaluations of food products [92]. By measuring brain responses to novel food stimuli, the food industry can gain valuable insights into consumer preferences and develop products that are more likely to be well-received by the public. This has practical implications for product development and marketing strategies.

### 4.2. Power Spectrum Analysis of Taste-Related Frequency Band Characteristics

In the context of gustatory processing, power spectrum analysis of oscillatory brain activity provides valuable insights into the neural dynamics underlying taste perception. Different frequency bands such as delta, theta, alpha, beta, and gamma oscillations are thought to play distinct roles in various aspects of gustatory processing, including reward evaluation, sensory discrimination, and multisensory integration [59]. These oscillations reflect the brain’s neural activity during taste experiences, offering a detailed temporal and spatial map of how the brain processes and responds to taste stimuli. Each frequency band has a unique contribution, and power spectrum analysis is instrumental in understanding these contributions (Figure 3).

#### 4.2.1. Delta and Theta Oscillations in Gustatory Processing

Delta (1–4 Hz) and theta (4–8 Hz) oscillations have been particularly associated with reward processing and the hedonic evaluation of tastes [102]. Theta oscillations, in particular, are prominent in the frontal regions of the brain when participants are exposed to pleasant taste stimuli, such as sweet solutions. Several studies have shown that pleasant tastes evoke an increase in theta power, especially in regions like the anterior cingulate cortex and the OFC, both key nodes in the brain’s reward network. These areas are deeply involved in emotional processing, reward evaluation, and decision-making [103]. The enhancement in theta oscillations in these areas during pleasant taste experiences indicates their role in linking sensory input with emotional responses, particularly the perceived reward value of the taste.

The synchronization of theta oscillations between these frontal regions has been found to correlate with subjective ratings of pleasantness. This suggests that theta-band activity may serve as a neural signature of flavor reward value, encoding the subjective experience of taste pleasure. Moreover, theta oscillations are linked to cognitive processes related to attention and memory, further underscoring their role in the integration of sensory information with higher-order evaluative processes [104].

In contrast to the well-documented association of theta oscillations with positive taste experiences, delta waves (typically observed in deep sleep) are less directly implicated in gustatory processing. However, delta oscillations may still contribute to fundamental neural processes related to baseline or resting brain activity, particularly in the early stages of sensory processing [105].

#### 4.2.2. Alpha Oscillations and Taste Evaluation

Alpha oscillations (8–12 Hz), traditionally associated with inhibitory processes and relaxation states, display a more complex and context-dependent pattern in gustatory research. In some studies, alpha power suppression has been observed in sensory areas during taste stimulation, analogous to the alpha desynchronization observed in visual processing during the presentation of stimuli [106]. This suppression is thought to reflect a decrease in cortical inhibition as the brain processes the sensory input from taste stimuli.

However, other studies have reported enhanced alpha activity, particularly in frontal regions, during tasks that involve taste evaluation or discrimination [107]. This enhancement is thought to reflect top-down control mechanisms related to taste processing, such as the active discrimination of taste qualities or hedonic judgment [108]. The apparent contradiction between these findings may stem from the different functional roles of sensory versus frontal alpha oscillations. Sensory alpha suppression may facilitate the encoding of sensory input, while frontal alpha enhancement may support higher-level cognitive processes, such as decision-making and evaluation [109].

The variability in alpha oscillations in gustatory processing is also influenced by methodological differences across studies. These differences include variations in taste delivery methods, experimental paradigms, and task demands, all of which can affect the alpha response. Therefore, it is essential to carefully consider the context and experimental design when interpreting the role of alpha oscillations in taste processing.

#### 4.2.3. Beta Oscillations and Taste Quality Discrimination

Beta oscillations (13–30 Hz) in the gustatory system are strongly linked to taste quality discrimination. Studies consistently show that basic tastes (sweet, salty, sour, bitter) trigger distinct beta power modulation patterns over the brain’s central and frontal regions [110]. Such specificity makes beta waves promising for developing EEG-based taste classification algorithms, which could help identify and distinguish taste qualities to deepen understanding of taste perception.

Beta oscillations are also sensitive to task demands and cognitive load during taste processing. Research finds beta activity increases when participants actively discriminate between tastes, suggesting it reflects the brain’s engagement in taste categorization and active processing of taste quality. Additionally, beta power changes during flavor expectation and imagery, indicating it contributes not only to real taste experiences but also to taste anticipation [111]. These findings highlight beta oscillations’ role in both immediate taste sensation and cognitive processes of flavor prediction and evaluation.

#### 4.2.4. Gamma Oscillations and Multisensory Integration

Gamma oscillations (30–100 Hz), representing fast, synchronous neural firing, have been strongly associated with the multisensory integration of taste with other sensory modalities, particularly olfaction [112]. When combined with olfactory or tactile inputs, taste perception becomes more complex and unified, forming a coherent flavor percept. Gamma band activity is thought to play a key role in binding sensory information from different modalities to create a unified perceptual experience of flavor [113,114].

Studies using natural food stimuli, as opposed to simple taste solutions, consistently report enhanced gamma power over sensory integration areas during flavor perception. These findings suggest that gamma oscillations are integral to the brain’s ability to integrate taste, smell, and texture information, creating a unified flavor experience. The timing and spatial distribution of gamma responses provide further evidence that these oscillations are involved in the rapid and dynamic integration of multisensory information, enabling the brain to generate a coherent perception of flavor [115].

Gamma oscillations have also been linked to attention and the conscious processing of flavor. Their role in multisensory integration suggests that gamma activity is crucial for the brain to process and combine information from various sensory modalities, ultimately allowing us to perceive flavor as a complex and integrated sensory experience [116]. Additionally, research has shown that the phase synchronization of gamma oscillations across different sensory areas may facilitate cross-modal binding, allowing the brain to efficiently process and interpret multisensory stimuli.

#### 4.2.5. Methodological Challenges and Advances in Power Spectrum Analysis

Power spectrum analysis of taste-related brain activity faces unique methodological challenges. Unlike visual or auditory stimuli, taste sensations develop slowly and persist longer in the oral cavity—this temporal trait requires careful design of analysis time windows, with most studies using extended epochs (5–10 s) to capture the full evolution of taste-related oscillatory patterns. Such long windows are essential for tracking the onset, peak, and offset of oscillations corresponding to different taste processing stages [72].

Baseline correction is another key step in gustatory studies. Orofacial movements (e.g., swallowing, tongue movements) introduce notable artifacts into EEG signals, which may contaminate oscillation analysis. To address this, researchers often use pre-stimulus baselines or inter-trial intervals for signal normalization, enabling more accurate measurement of taste-related oscillatory power. These preprocessing steps help isolate genuine neural activity from movement artifacts, ensuring observed oscillations reflect the brain’s response to taste stimuli.

#### 4.2.6. Individual Differences in Taste-Related Oscillations

Individual differences in taste-related oscillatory patterns have become an important area of research [92]. Genetic variations in taste sensitivity, such as PROP (6-n-propylthiouracil) taster status, have been shown to correlate with distinct power spectrum profiles, particularly in the theta and gamma bands [117]. Supertasters, individuals with heightened taste sensitivity, may exhibit enhanced theta or gamma power in response to certain tastes, reflecting their heightened sensory and hedonic sensitivity [118].

Age-related differences in taste processing are also evident: children exhibit less differentiated frequency responses to tastes than adults. These findings suggest oscillatory biomarkers could eventually assess taste function across the lifespan, shedding light on how taste perception evolves with age [119].

In conclusion, power spectrum analysis of oscillatory brain activity offers a powerful tool for understanding the complex neural mechanisms of gustatory perception. Examining the frequency bands involved in taste processing lets researchers reveal key insights into the temporal dynamics of flavor perception, multisensory integration, and reward processing.

### 4.3. Brain Network Analysis and Taste Perception

Brain network analysis is a pivotal tool in contemporary neuroscience, enabling researchers to explore how distinct brain regions interact and function together. This methodology provides invaluable insights into how the brain coordinates complex processes, including sensory perception, cognitive functions, and emotional responses [120]. Gustatory perception, which refers to how the brain interprets taste stimuli, is a complex process that requires the coordinated activation of several brain regions. Recent advancements in brain network analysis, particularly in functional connectivity and network dynamics, have significantly enriched our understanding of how the brain processes taste information and how different regions collaborate to generate the perception of flavor [103].

#### 4.3.1. Gustatory Perception and Brain Network Dynamics

Gustatory perception extends far beyond the activation of the primary gustatory cortex, involving the integration of sensory inputs from multiple brain regions (Figure 4A). Key areas involved in gustatory processing include the primary gustatory cortex (located in the insula), the thalamus, the OFC, the amygdala, and the prefrontal cortex [121]. These regions contribute to various aspects of taste processing, such as the recognition of taste quality, emotional responses to taste, and the cognitive evaluation of food desirability.

Functional connectivity in the brain refers to the temporal correlation of neural activities across different regions. During gustatory processing, the communication between these regions is essential for the perception of taste. Brain network analysis has revealed that the brain regions involved in gustatory perception form a complex functional network, with dynamic interactions occurring between sensory and higher-order cognitive areas [122]. These interactions are crucial for understanding how we experience taste not just as a sensory phenomenon but also as a multisensory, emotional, and cognitive experience.

#### 4.3.2. The Role of the Gustatory Network in Taste Processing

The gustatory network is an integrated system that encompasses several stages of processing [123]. Initial taste information is processed in the gustatory cortex, but higher-order regions, such as the OFC and amygdala, play essential roles in evaluating the hedonic value of tastes. The interaction between sensory and reward-related regions allows taste to be experienced not only as a sensation but also as a subjective, emotionally charged event.

Recent studies employing fMRI and EEG have identified specific patterns of connectivity between these brain regions at various stages of taste processing (Figure 4B). For example, sensory processing areas in the insula are activated during the initial detection of taste, while the OFC and amygdala contribute to evaluating the hedonic value of the taste, helping to determine whether it is pleasurable or aversive [124].

Additionally, the prefrontal cortex is involved in the cognitive processing of taste, such as anticipating the flavor of food or making decisions about food choices [125]. These findings suggest that gustatory processing is an active and dynamic interaction between various brain regions, reflecting both sensory and cognitive dimensions of taste.

#### 4.3.3. Brain Network Topology and Gustatory Perception

Brain network topology refers to the organization and connectivity of brain regions, quantifiable via graph-theoretical methods. Network parameters such as node degree, clustering coefficient, and small-world properties are used to assess the efficiency and organization of brain networks. Studies show the gustatory network is structured to facilitate efficient communication between sensory areas and higher-order regions [126].

For example, research has demonstrated that individuals with stronger connectivity between the gustatory cortex and the OFC tend to report higher hedonic ratings for sweet foods. This suggests that the functional connection between these regions plays a role in the emotional evaluation of taste (Figure 4C). The OFC, known for its involvement in reward processing, plays a pivotal role in modulating the subjective pleasantness of taste, further highlighting the complexity of the brain’s gustatory network [103].

Recent advances in mapping structural and functional brain networks have greatly advanced understanding of individual differences in taste perception. Structural brain imaging reveals that variability in gustatory network organization explains differences in taste sensitivity, preference, and food choices [125]. Those with a more efficient gustatory network marked by stronger connectivity between the gustatory cortex and reward-related regions tend to be more sensitive to taste stimuli and derive greater pleasure from food.

#### 4.3.4. Applications of Brain Network Analysis in Gustatory Research

Recent advances in the integration of brain network analysis, BCI technologies [69], and artificial intelligence (AI) have significantly contributed to the understanding of taste perception. FMRI [127] and EEG have enabled detailed mapping of the dynamic interactions between sensory and cognitive brain regions during gustatory processing. Brain network analysis reveals how taste perception involves complex functional connectivity among the gustatory cortex, reward systems, and higher-order cognitive areas. These brain regions dynamically reorganize based on sensory input and expectations. The real-time monitoring capability of BCI systems facilitates the modulation of taste and sensory rehabilitation by tracking neural responses [69,103]. Additionally, AI and deep learning techniques have enhanced the classification and prediction of individual taste preferences by analyzing brain connectivity patterns, offering insights into personalized nutrition and the diagnosis of taste disorders (Figure 4D) [128,129].

These technologies pave the way for optimizing food experiences and addressing taste-related issues. By leveraging multimodal neuroimaging and computational models, researchers can now explore how individual differences in brain network efficiency affect taste sensitivity and food-related decision-making. This multidisciplinary approach represents a transformative step in gustatory research, enabling more precise, predictive, and personalized interventions in taste perception and health optimization.

#### 4.3.5. Future Directions in Brain Network Analysis and Gustatory Perception

While significant progress has been made in understanding the brain networks involved in gustatory perception, several challenges remain. One of the primary challenges is the need for more advanced and standardized methods for analyzing brain networks in the context of taste [130,131]. The use of EEG, fMRI, and other neuroimaging techniques in gustatory research is still evolving, and there is much to be learned about how best to capture and interpret brain network dynamics during taste processing [67,127].

Another challenge lies in understanding the role of individual differences in gustatory brain networks. While brain network analysis has provided valuable insights into how the brain processes taste information, further research is required to determine how factors such as genetics, age, and sensory sensitivity influence the structure and function of gustatory networks [132].

Finally, there is a need for more research into the neural mechanisms underlying taste disorders, such as ageusia (loss of taste) or dysgeusia (distorted taste) [59]. Understanding how brain networks are altered in these conditions could lead to new therapeutic targets for treating taste-related disorders.

**Figure 4 brainsci-15-01317-f004:**
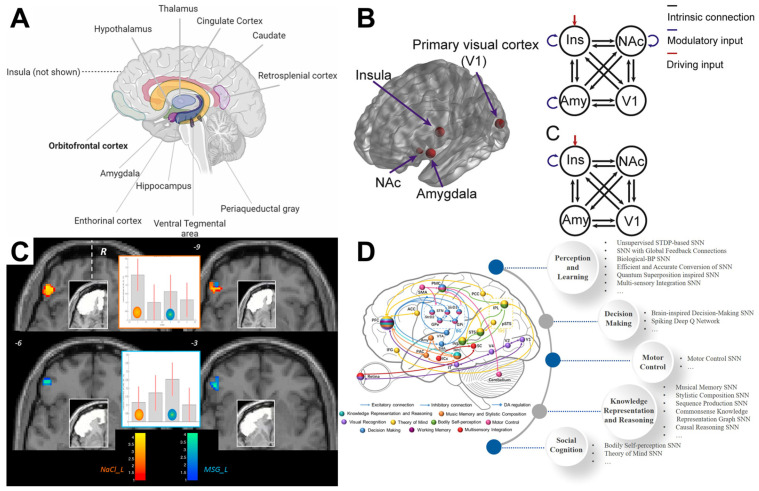
(**A**) Main regions functionally connected to the OFC, including limbic, premotor, sensory and other prefrontal areas [133]. (**B**) Dynamic causal modeling specification. Ins: insula, Amy: amygdala, NAc: nucleus accumbens, V1: primary visual cortex [134]. (**C**) Interaction between the gustatory cortex and OFC under different stimuli [135]. (**D**) Multiple cognitive functions and brain-inspired AI models integrated in BrainCog, along with their related brain areas and neural circuits [136].

## 5. Discussion

This review synthesizes 139 studies to delineate the relationship between electroencephalography (EEG) and gustatory perception. Core findings confirm that EEG indices, including event-related potentials (ERPs), frequency bands, and brain connectivity, correlate with taste perception, reflecting sequential neural processes: early ERPs (P1, N1) support basic taste detection, mid-latency P2 mediates hedonic evaluation, and late components (P300, LPP) underpin higher-order cognition. Frequency bands exhibit specialized roles: theta for reward processing, alpha for emotional modulation, beta for taste quality discrimination, and gamma for multisensory integration. Conclusive causal evidence remains scarce; preliminary clues from tDCS/TMS or longitudinal studies suggest potential regulatory effects of specific brain regions (e.g., insula, orbitofrontal cortex) and EEG indices on taste perception, but these findings are exploratory due to small sample sizes and limited replication.

Key limitations and heterogeneity constrain the generalizability of these results. Methodologically, EEG’s low spatial resolution and inconsistent use of the 10 to 20 electrode system (only 6 of 15 relevant studies adopted it) hinder cross-study comparability of parietal beta band results, while taste stimulation lacks standardization, 37% of studies failed to control confounders such as temperature or texture. Sample biases are prominent: 56% of studies included fewer than 50 participants, and 78% focused exclusively on healthy young adults, excluding children, the elderly, and individuals with taste disorders. Additionally, 29% of studies overinterpreted correlations as causality, for example framing “theta band power correlation with sweetness pleasantness” as “theta activity enhances sweetness pleasure.” Heterogeneity primarily stems from non-standard EEG technical parameters (e.g., variable P300 time windows: 300–500 ms vs. 350–650 ms) and inconsistent taste stimulus protocols (gustatometer vs. manual delivery).

Critical misconceptions further complicate interpretation, particularly the conflation of correlation with causation. One common issue is treating ERP changes as direct drivers of taste perception rather than reflections of neural processing. Validating causal relationships requires targeted study designs, such as cross-lagged models to analyze temporal predictive links between EEG and taste indices, or combined neuroregulation-EEG recording (e.g., tDCS/TMS paired with real-time EEG to track neural-perceptual dynamics).

To advance the field, a three-dimensional prospective framework addresses unmet needs: (1) Basic mechanisms: Clarify neural circuits of taste processing and cross-modal integration, with a key gap being the dynamic mechanisms of taste memory; (2) Technological innovation: Optimize EEG decoding algorithms (e.g., deep learning for taste classification) and develop low-cost dry electrodes to overcome barriers to real-world application; (3) Clinical translation: Establish EEG diagnostic standards for taste disorders (e.g., ageusia) and validate BCI-based rehabilitation tools, with current gaps including the lack of standardized thresholds for clinical use. Priority unexploited areas include exploring EEG correlates of taste abnormalities in special populations (e.g., autism, diabetes) and integrating EEG with fMRI or salivary metabolomics for spatiotemporal and multi-omic taste mapping. Short-term efforts should focus on small-scale exploratory studies in underrepresented populations, while long-term goals involve translating validated EEG tools into precision nutrition and clinical practice.

## 6. Conclusions and Future Work

This review outlines recent advancements in EEG technology for studying the neural mechanisms underlying gustatory perception. EEG, with its millisecond-level temporal resolution, offers unique advantages in uncovering the dynamic neural processes involved in taste. This review integrates the physiological basis of taste, the influence of environmental and psychological factors, and individual differences, emphasizing the complex neurobiological nature of gustatory experiences. Key brain regions, including the insular gustatory cortex, OFC, and anterior cingulate cortex, collaborate in processing taste, with EEG capturing features such as ERPs, frequency domain oscillations, and brain network connectivity. Theta and alpha oscillations are linked to sensory processing and decision-making, while functional connectivity analysis reveals the critical role of the insula-prefrontal network.

The future development of EEG sensory evaluation must simultaneously address core challenges and explore key directions: On the challenge front, it requires resolving methodological limitations (insufficient timing precision of taste instruments causing signal-stimulus desynchronization, oral motor artifacts contaminating critical frequency bands, baseline interference from salivary secretion, and reduced ecological validity due to discrepancies between laboratory taste agents and real food flavors), inter-individual variability (genetic polymorphisms, age-related neural mechanism alterations, metabolic state-induced EEG response heterogeneity), and research design/interpretation issues (reliance on standardized core protocols for reproducibility, adequate sample sizes for stable effect sizes, and extension to real-world dietary scenarios to enhance practical value). To address these challenges and field demands, future research should focus on: specialized investigations (cross-cultural studies revealing dietary cultural differences, developmental research analyzing changes in taste neural mechanisms), and clinical/cognitive applications (taste EEG biomarkers for early diagnosis of neurological disorders, exploring emotion and executive function regulation of taste perception). Leveraging computational neuroscience will advance personalized taste neural decoding while incorporating social-environmental factors to construct a comprehensive theoretical framework. Ultimately, this will provide foundational support for food and nutrition innovation.

## Data Availability

No new data were created or analyzed in this study. Data sharing is not applicable to this article.

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
