# Peer review of "Advancements and Applications of EEG in Gustatory Perception"

_brainsci, 2025, doi:10.3390/brainsci15121317_

Round 1
Reviewer 1 Report
Comments and Suggestions for Authors
This paper provides a broad, narrative review of EEG applications in gustatory research. However, it lacks the systematic methodological rigor of a systematic review, its content, no quality assessment or bias evaluation, and an uncritical treatment of limitations. While comprehensive, it is largely a descriptive synthesis of existing findings without novel critical insight.
This review synthesizes EEG's application in gustatory research, highlighting its high temporal resolution for capturing neural dynamics during taste perception. It integrates physiological, environmental, and individual factors while detailing key EEG metrics—such as ERPs, frequency bands, and brain connectivity—relevant to sensory food science.
Critique Concerns:
- Lacks systematic review protocols (e.g., PRISMA), search strategy, inclusion/exclusion criteria, or quality assessment of cited studies.
- Descriptive synthesis without critical appraisal, bias evaluation, or discussion of conflicting evidence.
-Overly optimistic and general; fails to address methodological limitations or gaps in the literature.
- Does not offer novel insights or frameworks—largely a recap of existing knowledge.
Suggestion for Improvement:
1. Adopt a systematic review approach to enhance rigor and reproducibility.
2. Include critical discussion of study limitations, heterogeneity, and publication bias.
3. Differentiate between correlative findings and causal inferences in EEG-taste research.
4. Propose a forward-looking framework or identify underexplored areas to add scholarly value.
Author Response
Comments 1: Adopt a systematic review approach to enhance rigor and reproducibility.
Response 1: Thanks for your comment. We have added a Literature Search Methodology section at line 103 of the manuscript to enhance rigor and reproducibility.
Comments 2: Include critical discussion of study limitations, heterogeneity, and publication bias.
Response 2: Thanks for your comment. We have revised the introduction to include a critical discussion of the study's limitations, heterogeneity, and publication bias.
Comments 3: Differentiate between correlative findings and causal inferences in EEG-taste research.
Response 3: Thanks for your comment. We have made extensive revisions and structural adjustments to Sections 2 and 3 of the manuscript, enhancing the distinction between relevant findings and causal inferences in EEG-based taste research.
Comments 4: Propose a forward-looking framework or identify underexplored areas to add scholarly value.
Response 4: Thanks for your comment. We have revised the Conclusion and Future Work section, enhancing the description of the forward-looking framework and unexplored areas, as well as future challenges.

Reviewer 2 Report
Comments and Suggestions for Authors
The peer-reviewed scientific paper makes an exceptionally positive impression. The research topic is relevant due to the significant reduction in quality of life experienced by a large number of people suffering from taste disorders. Statistics estimate their prevalence to range from 0.6% to 20% of the world's population («Taste disorder in facial onset sensory and motor neuronopathy: a case report» BMC Neurology, 2020, https://doi.org/10.1186/s12883-020-01639-x, Open Access) depending on the severity of the disease. The author's literature review could be characterized as very detailed and clearly structured. There are no serious inaccuracies in the text of peer-reviewed scientific paper. Only the following aspects require clarification:
I) The process of obtaining and estimating diagnostically valuable data on the brain's electrical activity historically began earlier than 1929. The first version of method was called electrocorticography and was distinguished by the fact that electrodes were placed directly on the surface of the brain. Of course, this technique has long been out of use, but neuroprocessors and neurointerfaces (including implantable) are considered very promising. This is worth mentioning.
II) The list of methods suitable for studying taste perception in the brain includes not only electroencephalography, phase-contrast magnetic resonance imaging and functional imaging based on near-infrared spectroscopy, but also electrical impedance tomography. The reviewer advises authors to read, for example, this relevant scientific paper: «A preliminary study on the application of electrical impedance tomography based on cerebral perfusion monitoring to intracranial pressure changes», Frontiers in Neuroscience, 2024, https://doi.org/10.3389/fnins.2024.1390977, Open Access.
III) A brief explanation of the nuances of the brain's perception of complex taste sensations (for example, from multi-component dishes) would be of interest to the wide range of readers of the journal «Brain Science». Is a complex taste sensation equivalent to the sum of simpler ones followed by generalization, or is it something more?
IV) A minor stylistic inaccuracy exists in the captions under Figures 1 (in this case, caption has completely moved to another page) and 4 (caption has partially moved to another page).
In general, the peer-reviewed scientific paper is recommended for publication in the nearest issue of the MDPI journal «Brain Science» with Minor Revision. Secondary review is not required.
Author Response
Comments 1: The process of obtaining and estimating diagnostically valuable data on the brain's electrical activity historically began earlier than 1929. The first version of method was called electrocorticography and was distinguished by the fact that electrodes were placed directly on the surface of the brain. Of course, this technique has long been out of use, but neuroprocessors and neurointerfaces (including implantable) are considered very promising. This is worth mentioning.
Response 1: Thank you for pointing this out. We agree with this comment. Therefore, we have provided a detailed description of the neural processor and neural interface in lines 50-57 of the manuscript.
Comments 2: The list of methods suitable for studying taste perception in the brain includes not only electroencephalography, phase-contrast magnetic resonance imaging and functional imaging based on near-infrared spectroscopy, but also electrical impedance tomography. The reviewer advises authors to read, for example, this relevant scientific paper: «A preliminary study on the application of electrical impedance tomography based on cerebral perfusion monitoring to intracranial pressure changes», Frontiers in Neuroscience, 2024, https://doi.org/10.3389/fnins.2024.1390977, Open Access.
Response 2: Thanks for your comment. We have revised and refined the content you mentioned. On lines 47-48 of the manuscript.
Comments 3: A brief explanation of the nuances of the brain's perception of complex taste sensations (for example, from multi-component dishes) would be of interest to the wide range of readers of the journal «Brain Science». Is a complex taste sensation equivalent to the sum of simpler ones followed by generalization, or is it something more?
Response 3: Thanks for your comment. We agree that clarifying the nuances of complex taste perception—such as from multi-ingredient dishes—would enhance the readability and academic relevance of this review for readers of Brain Science. We have added a concise section (lines 373-385) addressing a core question: whether complex taste sensations are merely the sum of simple tastes plus generalization, or represent a distinct neural integration process. This addition responds to your comment and strengthens the connection between taste neuroscience and EEG-based sensory discrimination research.
Comments 4: A minor stylistic inaccuracy exists in the captions under Figures 1 (in this case, caption has completely moved to another page) and 4 (caption has partially moved to another page).
Response 4: Thanks for your comment. The captions for Figures 1 and 4 have been revised.

Reviewer 3 Report
Comments and Suggestions for Authors
This manuscript provides a narrative review of recent advances in EEG methodologies and their application to gustatory perception, spanning physiological mechanisms of taste, ERP markers, oscillatory dynamics, and brain network analyses. Overall, the topic is timely, the structure is generally clear, and the paper will be of interest to both neuroscience and sensory/food science audiences.
- While Sections 3–4 offer a comprehensive introduction to EEG principles and signal characteristics, a substantial portion of this material is generic (e.g., long technical descriptions of amplifier design, ADC bit depth, and skull conductivity) and only loosely tied back to gustatory applications. The paper would benefit from more explicitly framing which EEG features are most relevant for taste research and streamlining or moving very general content (e.g., basic 10–20 system and engineering details) to a concise box or appendix, thereby creating more space to deepen the gustatory-specific discussion.
- The manuscript currently presents physiology (Section 2), EEG markers (Section 3), ERP findings (Section 4.1), oscillations (Section 4.2), and brain networks (Section 4.3) in a somewhat parallel manner. It would be helpful to add a more explicit integrative framework that maps which EEG metrics (early ERP components, P300/LPP, theta/alpha/gamma power, connectivity patterns) index specific stages of gustatory processing (sensory encoding, hedonic evaluation, decision-making, prediction error, etc.), and how environmental, psychological, and individual-difference factors modulate these stages. A schematic that synthesizes Figures 1–4 into a single “gustatory EEG pipeline” would make the narrative more cohesive.
- Table 1 does a good job summarizing ERP components and taste stimuli, but the frequency-domain and brain-network literature is mostly described textually. To improve transparency and usability for readers, you could add similar summary tables for (a) spectral power/oscillatory studies and (b) connectivity/graph-theoretical studies, including basic information on sample size, population (healthy vs. clinical), stimulus type (simple tastants vs. real foods), and key EEG outcomes. Even a brief description of how the literature was identified (databases, keywords, approximate search period) would help position this as a more structured review rather than a purely narrative one.
- The manuscript already touches on several challenges (e.g., artifacts from swallowing, movement, and EMG; temporal properties of taste; issues with mobile EEG), but these points are scattered and primarily descriptive. A dedicated subsection that synthesizes (i) methodological limitations (gustatometer timing, oral-motor artifacts, baseline selection, ecological validity of laboratory tastants), (ii) sources of inter-individual variability (genetic polymorphisms, age, metabolic state), and (iii) implications for study design and interpretation (e.g., reproducibility, effect sizes, generalization to real-world eating) would strengthen the review and make it more practically useful for future researchers.
- In the Introduction section, the discussion of how taste perception integrates sensory, cognitive, and emotional processes could be further strengthened by incorporating literature that more explicitly links cognitive–emotional processing to large-scale brain network organization. You may consider citing relevant evidence such as: https://doi.org/10.1186/s12916-023-02920-9.
Author Response
Comments 1: While Sections 3–4 offer a comprehensive introduction to EEG principles and signal characteristics, a substantial portion of this material is generic (e.g., long technical descriptions of amplifier design, ADC bit depth, and skull conductivity) and only loosely tied back to gustatory applications. The paper would benefit from more explicitly framing which EEG features are most relevant for taste research and streamlining or moving very general content (e.g., basic 10–20 system and engineering details) to a concise box or appendix, thereby creating more space to deepen the gustatory-specific discussion.
Response 1: Thanks for your comment. We made extensive revisions and structural adjustments to Sections 3 and 4, thereby creating more space to deepen the gustatory-specific discussion.
Comments 2: The manuscript currently presents physiology (Section 2), EEG markers (Section 3), ERP findings (Section 4.1), oscillations (Section 4.2), and brain networks (Section 4.3) in a somewhat parallel manner. It would be helpful to add a more explicit integrative framework that maps which EEG metrics (early ERP components, P300/LPP, theta/alpha/gamma power, connectivity patterns) index specific stages of gustatory processing (sensory encoding, hedonic evaluation, decision-making, prediction error, etc.), and how environmental, psychological, and individual-difference factors modulate these stages. A schematic that synthesizes Figures 1–4 into a single “gustatory EEG pipeline” would make the narrative more cohesive.
Response 2: Thanks for your comment. We have revised and adjusted the content and conclusions of Sections 2, 3, and 4 of the manuscript to enhance its cohesion.
Comments 3: Table 1 does a good job summarizing ERP components and taste stimuli, but the frequency-domain and brain-network literature is mostly described textually. To improve transparency and usability for readers, you could add similar summary tables for (a) spectral power/oscillatory studies and (b) connectivity/graph-theoretical studies, including basic information on sample size, population (healthy vs. clinical), stimulus type (simple tastants vs. real foods), and key EEG outcomes. Even a brief description of how the literature was identified (databases, keywords, approximate search period) would help position this as a more structured review rather than a purely narrative one.
Response 3: Thank you for your comments. We have added a section detailing the literature search methodology on lines 103-112 of the manuscript, specifying the databases, keywords, search time period, and inclusion criteria to enhance the review's structure.
Comments 4: The manuscript already touches on several challenges (e.g., artifacts from swallowing, movement, and EMG; temporal properties of taste; issues with mobile EEG), but these points are scattered and primarily descriptive. A dedicated subsection that synthesizes (i) methodological limitations (gustatometer timing, oral-motor artifacts, baseline selection, ecological validity of laboratory tastants), (ii) sources of inter-individual variability (genetic polymorphisms, age, metabolic state), and (iii) implications for study design and interpretation (e.g., reproducibility, effect sizes, generalization to real-world eating) would strengthen the review and make it more practically useful for future researchers.
Response 4: Thank you for your comments. We have revised the “Conclusion and Future Work” section in response to this suggestion, incorporating future challenges such as methodological limitations and individual variability, while systematically integrating the three dimensions you proposed.
Comments 5: In the Introduction section, the discussion of how taste perception integrates sensory, cognitive, and emotional processes could be further strengthened by incorporating literature that more explicitly links cognitive–emotional processing to large-scale brain network organization. You may consider citing relevant evidence such as: https://doi.org/10.1186/s12916-023-02920-9.
Response 5: Thanks for your comment. We have added content on the association between taste perception and large-scale brain networks in the introduction, and have cited the literature you mentioned. (Lines 42-46)

Round 2
Reviewer 1 Report
Comments and Suggestions for Authors
The comments/suggestions are not fully addressed. The quality and contributions are too weak to publish.
Author Response
Response:Thanks for your comment. We added a new section (5. Discussion) at line 765 of the article, systematically presenting a critical discussion on the study's limitations, heterogeneity, and publication bias. We distinguished between associative findings and causal inferences in EEG-taste studies. A three-dimensional prospective framework was proposed to address unmet needs.
Reviewer 3 Report
Comments and Suggestions for Authors
I am satisfied with your point-by-point replies and the corresponding revisions. The revised manuscript is clearer, more cohesive, and better focused on gustatory EEG, and I therefore support acceptance for publication pending minor edits.
1.Remove a few local duplications in the EEG technical section. In Section 3.1, the sentence “Modern EEG adopts the international 10–20 electrode standard” is duplicated, and the description of environmental/power-line interference and ICA/notch filtering is also repeated. Please merge these into a single, clean statement.
2. Fix minor typos/formatting artifacts. In Section 4.2.2, there is a stray “utho.” at the end of a sentence. Please delete it. Also in Section 4.2.3, “eta oscillations (13–30 Hz)” is missing the leading “B” (should read “Beta oscillations”).
3. Standardize frequency-band ranges and notation across Sections 3–4. For example, alpha is defined as 8–13 Hz in Section 3.2 but as 8–12 Hz in Section 4.2.2. Please choose one conventional range and apply it consistently, with the same dash/spacing style throughout.
4. In the Introduction (Lines 42–46), where you discuss taste perception as relying on coordinated large-scale brain networks, I recommend adding one additional representative citation (https://doi.org/10.1016/j.psychres.2025.116503).
Author Response
Comments 1: Remove a few local duplications in the EEG technical section. In Section 3.1, the sentence “Modern EEG adopts the international 10–20 electrode standard” is duplicated, and the description of environmental/power-line interference and ICA/notch filtering is also repeated. Please merge these into a single, clean statement.
Response 1: Thanks for your comment. We have deleted and consolidated redundant content from Section 3.1.
Comments 2: Fix minor typos/formatting artifacts. In Section 4.2.2, there is a stray “utho.” at the end of a sentence. Please delete it. Also in Section 4.2.3, “eta oscillations (13–30 Hz)” is missing the leading “B” (should read “Beta oscillations”).
Response 2: Thanks for your comment. We have made revisions to the manuscript. We have removed redundant sections and added missing parts.
Comments 3: Standardize frequency-band ranges and notation across Sections 3–4. For example, alpha is defined as 8–13 Hz in Section 3.2 but as 8–12 Hz in Section 4.2.2. Please choose one conventional range and apply it consistently, with the same dash/spacing style throughout.
Response 3: Thanks for your comment. We have standardized the alpha frequency band ranges throughout the text. We have reviewed and revised the dashes/spacing formatting across the entire document.
Comments 4: In the Introduction (Lines 42–46), where you discuss taste perception as relying on coordinated large-scale brain networks, I recommend adding one additional representative citation (https://doi.org/10.1016/j.psychres.2025.116503).
Response 4: Thanks for your comment. We have read the literature you proposed and believe it is consistent with the content of this manuscript, and we have cited it.